# Absorptive pinhole collimators for ballistic Dirac fermions in graphene

Arthur W. Barnard[1], Alex Hughes[1], Aaron L. Sharpe[2], Kenji Watanabe[3], Takashi Taniguchi[3] & David Goldhaber-Gordon[1]

Ballistic electrons in solids can have mean free paths far larger than the smallest features patterned by lithography. This has allowed development and study of solid-state electron-optical devices such as beam splitters and quantum point contacts, which have informed our understanding of electron flow and interactions. Recently, high-mobility graphene has emerged as an ideal two-dimensional semimetal that hosts unique chiral electron-optical effects due to its honeycomb crystalline lattice. However, this chiral transport prevents the simple use of electrostatic gates to define electron-optical devices in graphene. Here we present a method of creating highly collimated electron beams in graphene based on collinear pairs of slits, with absorptive sidewalls between the slits. By this method, we achieve beams with angular width 18° or narrower, and transmission matching classical ballistic predictions.

[1] Department of Physics, Stanford University, Stanford, California 94305, USA. [2] Department of Applied Physics, Stanford University, Stanford, California 94305, USA. [3] National Institute for Materials Science, 1-1 Namiki, Tsukuba 305-0044, Japan. Correspondence and requests for materials should be addressed to A.W.B. (email: barnarda@stanford.edu) or to D.G.-G. (email: goldhaber-gordon@stanford.edu).

In the absence of scattering, electrons propagate freely as coherent waves, analogous to light in free space. Capitalizing on this behaviour, electron-optical elements including beam splitters[1,2], quantum point contacts[3,4], lenses[5], wave guides[6,7] and mirrors[8] have been fashioned in solid-state two-dimensional electron systems[9] (2DESs). The 2DES in graphene hosts chiral electrons[10–14], with unique refractive properties and associated novel opportunities for electron optics[12,13,15,16]. Until recently, disorder-induced scattering has limited implementation of these ideas. Encapsulation of graphene in hexagonal boron nitride (hBN)[17,18] now enables striking manifestations of refractive ballistic transport[15] including quasiparticle dynamics in superlattices[19], snake states[20] and Veselago lenses[21]. A collimated electron source could be the final piece needed to unlock the potential of electron refraction in graphene, enabling diverse applications such as ballistic transistors[22,23], flying qubits[24] and electron interferometers[25]. In conventional semiconductor 2DESs, electrons can be collimated by quantum point contacts[3] to form narrow beams. In graphene, however, electrons are not readily confined by gates and alternative proposals[26–28] for collimation in graphene have yet to be realized.

Here we demonstrate experimentally and validate computationally an electron collimator based on a collinear pair of pinhole slits in hBN-encapsulated graphene. We show that grounded edge contacts[17]—analogous to peripheral surfaces painted black in an optical system—can efficiently remove stray electron trajectories that do not directly traverse the two pinholes, leaving a geometrically defined collimated beams.

## Results

**Collimator design and function.** An absorptive pinhole collimator is constructed from an etched graphene heterostructure with a two-chamber geometry wherein independent electrodes make ohmic contact to each chamber (Fig. 1a). The contact to the bottom chamber (red, Fig. 1a) serves as the source for charge carriers, while the contact to the top chamber (black, Fig. 1a) acts as an absorptive filter. To realize a collimating configuration, the filter contact (F) is grounded and the source contact (S) is current biased; charge carriers are isotropically injected from the source, but only those trajectories that pass through both pinhole apertures reach the graphene bulk. Applying a uniform magnetic field can steer the collimated beam. For an uncollimated configuration, the filter and source contacts are electrically shorted.

Our device consists of hBN-encapsulated graphene etched into a Hall-bar-like geometry with the voltage probes replaced by collimating contacts (Fig. 1b). The hBN layers are both $d_{BN} \sim 80$ nm thick and the device is assembled on $d_{ox} = 300$ nm SiO$_2$ atop a degenerately doped silicon substrate used as a back gate to tune charge carrier density $n$. To test the collimation behaviour of an individual injector in the ballistic regime, we perform a non-local magnetotransport measurement, injecting from one collimator and probing trajectories that reach across the width of the device ($W_{dev} = 2\,\mu$m) in the collimated and uncollimated configurations (green and blue respectively, Fig. 1c). We inject from the lower right collimator (labelled S4,F4) throughout this Article and, in this case, measure the voltage of the upper right collimator (labelled S3,F3) relative to a reference (F1). In the presence of a B-field, electron trajectories that pass from the injector to collector flow from the injector at an angle $\theta = \sin^{-1}\frac{qBW_{dev}}{2\hbar\sqrt{n\pi}}$, where $q$ is the quasiparticle charge. From this, we find that the angular full width at half maximum (FWHM) is 70° when injecting in the uncollimated configuration and 18° when injecting in the collimated configuration.

For an uncollimated source[3], the angular conductance is expected to go as $G(\theta) = \frac{2e^2}{h}\sqrt{\frac{n}{\pi}}w_0\cos(\theta)$, where $\frac{2e^2}{h}\sqrt{\frac{n}{\pi}}$ is the flux density at the Fermi level and $w_0\cos(\theta)$ is the projected width of the contact. The collector has an acceptance angle of $\frac{w_0}{W_{dev}}\cos(\theta)$, leading to an expected $\cos^2(\theta)$ distribution ($\theta_{FWHM} = 90°$). The 70° FWHM for our uncollimated data is in reasonable agreement with this expectation given that the reference contact collects more electrons at higher B-fields and thus suppresses the signal at high angles.

In our collimators, the flux density at the Fermi level is identical to that in a single slit, but the projected width is geometrically defined by the pinhole width $w_0$ and pinhole separation $L_0$. For small angles $|\theta| < \tan^{-1}w_0/L_0$, the projected width $w(\theta) = \cos(\theta)[w_0 - L_0|\tan(\theta)|]$ (left, Fig. 1d). At larger angles, no carriers should transmit, yielding:

$$G(\theta) = \frac{2e^2}{h}\sqrt{\frac{n}{\pi}}\cos(\theta)[w_0 - L_0|\tan(\theta)|]; |\theta| < \tan^{-1}\frac{w_0}{L_0}. \quad (1)$$

Convolving over the acceptance angle of the collector (see Supplementary Note 1 for details), we calculate the angular conductance distribution (middle, Fig. 1d) for both the uncollimated case (blue) and the collimated case (green) with $w_0 = 300$ nm and $L_0 = 850$ nm, consistent with the fabricated collimator dimensions. The FWHM of the collimator emission is 22° for theory and 18° for experiment (right, Fig. 1d), showing that our injectors efficiently filter wide-angle trajectories and transmit narrowly collimated beams.

**Conductance of collimators.** Having established that the angular distribution of injected charge carriers is well described by classical ballistic theory, we now measure our collimators' conductance to determine how efficiently electrons traverse the pinholes. For this, we bias the injector in the collimating configuration (F4 grounded) and measure the current reaching all remaining electrodes as a function of gate voltage (Fig. 2a). The conductance of the collimator tunes sublinearly with $n$: $G \sim \sqrt{n - n_0}$ (dotted line, Fig. 2a). This qualitatively agrees with ballistic expectations ($G \sim \sqrt{n}$): integrating equation (1) over all angles, we expect:

$$G = \frac{4e^2}{h}\sqrt{n/\pi}\left[\sqrt{L_0^2 + w_0^2} - L_0\right]. \quad (2)$$

The small offset $n_0 \sim 1.6 \times 10^{11}$ cm$^{-2}$ in our measurement appears to result from diffraction by collimator slits (Fig. 2a, see Supplementary Notes 3 and 4 for details). Comparing equation (2), with the fit in Fig. 2a assuming $n \gg n_0$), indicates a conductance that is 35% of expectations. This is a lower bound for the transmission probability, because the collimating filter (F4) can reabsorb electrons that have diffusely scattered off of device edges.

To understand the impact of diffuse scattering and better estimate the transmission probability, we measure the current collected at specific detectors as a function of B-field. Having sourced $I_{source} = 50$ nA $< \frac{k_B T}{eR_{source}}$ (see Supplementary Note 6 for details), we collect current in detectors collinear with (red and blue, Fig. 2b) and adjacent to (black, Fig. 2b) the injector. Current collected at the collinear detector with a wide acceptance angle (red) peaks near $B = 0$, as the collimated beam travels straight across the device. The apparent background current is $\sim 3$–5% of $I_{source}$. At $B \sim 120$ mT, ballistic cyclotron orbits instead reach the adjacent detector, leading to a prominent peak in current detected at S1 (black) with F1 grounded. Coincident with this peak, the diffuse background of the collinear detector dips, as ballistic trajectories are consumed by the adjacent detector, reducing the number of electrons that eventually find their way into the collinear detector.

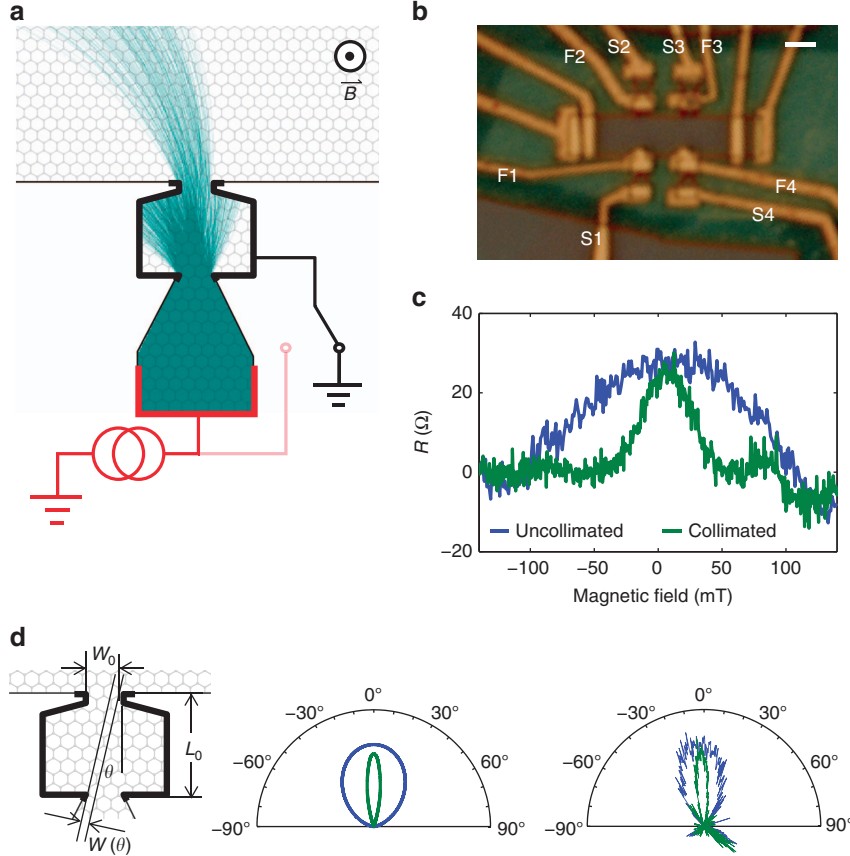

**Figure 1 | Absorptive pinhole collimators.** (**a**) Double pinhole collimator schematic. Current is sourced from bottom contact (red), passes through the bottom aperture and is either absorbed by the top contact (black) or passes into the device bulk. Only trajectories that pass through both apertures reach the bulk, producing a collimated beam. The collimated beam is steered by an external $B$-field. (**b**) Optical micrograph of device with four collimators in a Hall-bar-like geometry. Scale bar, $2\,\mu m$. (**c**) Measuring angular distribution. Non-local resistance at $n = 1.65 \times 10^{12}\,cm^{-2}$ (Fermi wavelength: $\lambda_f = 27.6\,nm$) is plotted with $V_{S3F3}$ measured relative to $V_{F1}$ when current is sourced from both S4 and F4 (blue), and only from S4, whereas F4 is grounded (green). The narrowness of the central peak for the F4-grounded data results from collimation. (**d**) Theoretical collimation behaviour versus experiment. Left: diagram of effective collimator width $w(\theta)$ at a fixed angle for classical ballistic trajectories. Middle: polar plot of theoretical angular dependence for a 300 nm-wide point contact (blue) and a $w_0 = 300\,nm$, $L_0 = 850\,nm$ collimator (green). Right: experimental data from **c** mapped to angle.

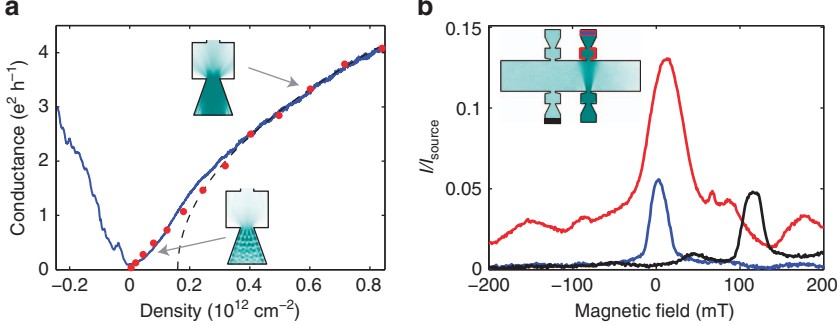

**Figure 2 | Conductance of single and paired collimators.** (**a**) Conductance of collimator measured in a three terminal configuration: S4 is current-biased, F4 is grounded and all remaining terminals are measured with a single current amplifier. The measured conductance (blue) scales as $\sqrt{n - n_0}$ (black dotted line), qualitatively agreeing with ballistic conduction of bulk graphene. Numerical solutions to the 2D Dirac equation (red dots) account well for low-density effects associated with diffraction. (**b**) Conductance measurements through angularly sensitive collectors. Current is collected at F3 + S3 (red), S3 (blue) and S1 (black) with all remaining contacts grounded. F3 + S3 has a broad background due to diffuse edge scattering and imperfect ohmic contacts. S3 has a FWHM of 8.5° due to double collimation and has minimal diffuse background. The peak height of S3 indicates nearly perfect ballistic transmission.

In light of the non-trivial diffuse background, we measure current with a narrow acceptance angle at the collector, rejecting most scattered electrons and thus better determining the transmission probability of the collimator. The resulting doubly collimated beam (blue) has a FWHM of 8.5°. Together, all these collinear apertures act as a single collimator with $L_0 = 3,750\,nm$ (the separation between the farthest-apart apertures). All of the injected current passes through the first aperture, so the fractional

current collected should be $\frac{G(w_0=300\,\text{nm},\,L_0=3,750\,\text{nm})}{G(w_0=300\,\text{nm},\,L_0=0)}=0.040$. The maximum of the doubly collimated peak is 0.056 (Fig. 2b). Subtracting a background of 0.005–0.015 (see Supplementary Note 5 for details) suggests transmission through the full path is $1.18\pm0.12$ times the expected value. The 20% beamwidth narrowing observed above for a single collimator (18° versus 22° expected) may indicate modest focusing, which would be consistent with slightly enhanced transmission through the double collimator. The excellent quantitative agreement shows that charge carriers transmit nearly perfectly from slit to slit. By demonstrating not only narrow beams but also high transmission probabilities, our measurements show that absorptive pin-hole filtering could produce low-noise, coherent, collimated beams of electrons in 2DESs that cannot be depleted by electrostatic gating.

**Transverse electron focusing.** Having experimentally demonstrated that absorptive pinhole collimators can controllably emit electron beams in hBN-encapsulated graphene heterostructures, we illustrate our technology's utility by aiming a beam at the edges of our graphene device to learn about the low-energy scattering behaviour of etched edges in these heterostructures. We perform three simultaneous non-local resistance measurements (Fig. 3a) to probe the specularity of reflections off various edges of the device. In Fig. 3b, we map $R_1 = \frac{V_1}{I_{\text{in}}}$ as a function of B-field and electron density. In both the electron-doped and hole-doped regimes, a peak near $B=0$ corresponds to ballistic quasiparticles being collected by the collinear contact in the absence of magnetic

deflection. Peaks in $R_1$ also appear at higher fields, primarily in the hole-doped regime ($n<0$). For reference, we plot contours corresponding to cyclotron radius $r=W/2$. Any features outside the parabolas ($r<W/2$) cannot correspond to direct ballistic quasiparticle transport across the width of the device and must involve scattering. These data imply that holes undergo multiple reflections at high B-fields, suggesting that the edges may scatter more specularly when hole-doped than when electron-doped.

To directly probe the specularity of reflections in our device, we perform a collimated transverse-electron focusing (TEF) measurement[8,29]. Probe $V_3$ at the lower left detector is even more sensitive than traditional TEF measurements to scattering that modifies ballistic trajectories, as here the injector and detector have narrow emission and acceptance angles, respectively. $R_3 = \frac{V_3}{I_{\text{in}}}$ as a function of electron density and B-field has several distinct features associated with specific cyclotron radii (Fig. 3c), in particular for hole doping. At $r_1=1.25\,\mu\text{m}$, there is a sharp peak with a FWHM of ~300 nm in both the hole and electron regimes. Although a conventional TEF peak would occur at $r_1 = \frac{L_{\text{lat}}}{2}$, where $L_{\text{lat}}=2.3\,\mu\text{m}$ is the lateral separation of injector and detector, our measured peak corresponds to slightly greater cyclotron radius. This is expected for our collimator geometry: we illustrate the expected $r_1$ trajectory in Fig. 3a and plot its corresponding contour in Fig. 3c, indicating excellent agreement with our measurement (see also Supplementary Note 2 for calculation). Trajectories at $r_1$ are insensitive to edge scattering, whereas at smaller $r$ (larger B) additional peaks imply specular reflection. In the electron-doped regime the presence of a

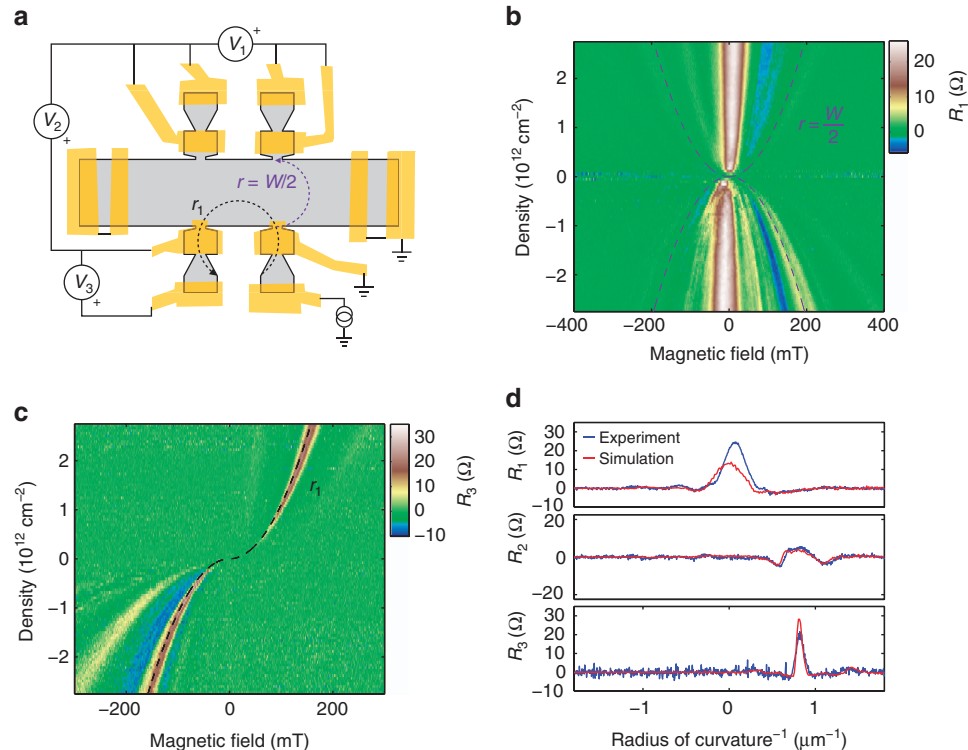

**Figure 3 | Probing edge scattering.** (**a**) Non-local resistance measurement schematic. (**b**) Resistance map characterizing angular profile of injected trajectories. A central peak near $B=0$ corresponds to the beam passing straight across the width of the device (a small angular offset is due to fabrication imperfections). The remainder of the electron-doped regime ($n>0$) is nearly featureless, whereas the hole-doped regime ($n<0$) has several auxiliary peaks. Dotted lines correspond to cyclotron orbits with radius equal to the half of device width ($r=W_{\text{dev}}/2$); features outside the two parabolas cannot correspond to direct ballistic trajectories between injector and collector. (**c**) Collimated transverse electron focusing. A sharp feature at $r_1=1.25\,\mu\text{m}$ corresponds to trajectories that pass through four pinholes. Features at higher magnetic field must involve specular reflections off of the device edge. There is no such feature on the electron side, whereas there is a noticeable band on the hole side. (**d**) Comparison of experimental data with classical ballistic simulation. Experimental data (blue) are taken at $n=2.7\times10^{12}\,\text{cm}^{-2}$ and simulation (red) assumes fully diffuse edge scattering and 67% ohmic transmission.

prominent peak at $r_1$ with no appreciable secondary peak suggests completely diffuse scattering, whereas in the hole-doped regime the presence of a significant secondary peak suggests appreciable specular reflection.

To validate this understanding and quantitatively determine the degree of specularity, we next carry out device-scale simulations of ballistic trajectories—treating electrons as classical point-like particles is warranted given that the Fermi wavelength $\lambda_f$ is in most cases much smaller than geometric features in our device and given that most trajectories are captured in ohmic contacts before having a chance to interfere. Modelling the fabricated device geometry including all ohmic contacts, we simulate electron emission from the injector, allowing for reflection off edges and interaction with floating or grounded ohmics. With two free parameters, transmission of ohmics $p_{trans}$ and probability of diffuse edge scattering $p_{diffuse}$, we simulate the measurement configuration shown in Fig. 3a–c (see Supplementary Note 7 for simulation details). The striking similarities between simulation ($p_{trans} = 67\%$ and $p_{scatter} = 100\%$) and measurement suggest that edge scattering is diffuse in our device in the electron-doped regime (Fig. 3d, see Supplementary Movie 1 for visualizing a $B$-sweep). Similar analysis yields $p_{trans} = 10\%$ and $p_{scatter} = 67\%$ in the hole-doped regime, quantitatively demonstrating significant electron-hole asymmetry in both ohmic contact properties and specularity of edge scattering in our device. This asymmetry may occur due to finite edge doping that induces smooth electrostatic edge barriers[30] in the $p$-doped regime.

## Discussion

The strong agreement between theory and experiment for both individual collimators and our entire collimating device indicates that absorptive collimation in high-mobility graphene devices can be predictably and robustly applied in a variety of geometries, opening the door for scientific and technological use of narrow electron beams in 2DESs. For example, Klein tunnelling[12,13,31] and Andreev reflections[32] are highly angularly dependent phenomena whose experimental signatures are obscured in typical transport experiments. In such cases, collimation-based measurements will illuminate the physics by quantitatively testing transmission and reflection at specific angles rather than integrated over a range of angles as in past experiments. In addition, novel technologies such as ballistic magnetometers may be built on the sharp magnetotransport features we achieve. Collimated sources are an important addition to the growing toolbox of electron-optical elements in ballistic graphene devices that enable a new class of transport measurements.

## Methods

**Sample fabrication.** Flakes of graphene (from highly oriented pyrolytic graphite, Momentive Performance Materials ZYA grade) and of hBN (from single crystals grown by high-pressure synthesis) were prepared[17] by exfoliation (3M Scotch 600 Transparent Tape) under ambient conditions (35–60% relative humidity) on $n$-doped silicon wafers with 90 nm thermal oxide (WRS Materials). The heterostructure was assembled by a top–down dry pick-up technique[19]. The completed heterostructure was deposited on a chip of $n^{++}$-doped silicon with 300 nm thermal oxide (WRS Materials). Polymer residue from the transfer process was removed by annealing the sample in a tube furnace for 1 h at 500 °C under continuous flow of oxygen (50 s.c.c.m.) and argon (500 s.c.c.m.)[33]. Device patterns were defined by e-beam lithography and reactive ion etching[19]. Ohmic contacts were established to the device using electron-beam evaporated Cr/Au electrodes to the exposed graphene edge[17].

**Measurement.** All measurements were performed at 1.6 K in the vapour space of a He flow cryostat with a superconducting magnet. Lock-ins (Stanford Research Systems SR830) at 17.76 Hz were used in all measurements; voltages were measured with Stanford Research Systems SR 560 voltage preamplifiers and currents were measured with Ithaco 1,211 current preamplifiers. The charge density $n$ was calculated from Shubnikov-de-Haas oscillations

$\left(\frac{n}{V_g} = 5.51 \times 10^{10}\, \text{cm}^{-2}\text{V}^{-1}\right)$, in good agreement with the expected geometric capacitance.

**Data availability.** The data sets generated during and/or analysed during the current study are available from the corresponding authors on reasonable request.

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

## Acknowledgements

We thank M. Lee and T. Petach for fruitful discussions. This work was financially supported by the Gordon and Betty Moore Foundation through Grant GBMF3429, by a Nano- and Quantum Science and Engineering Postdoctoral Fellowship (A.W.B.), by a Ford Foundation Predoctoral Fellowship (A.L.S.) and a National Science Foundation Graduate Research Fellowship (A.L.S.). K.W. and T.T. acknowledge support from the Elemental Strategy Initiative conducted by the MEXT (Japan). T.T. acknowledges support from JSPS Grant-in-Aid for Scientific Research under grants 262480621 and 25106006. Part of this work was performed at the Stanford Nano Shared Facilities (SNSF) supported by the National Science Foundation under award ECCS-1542152

## Author contributions

A.W.B., D.G.-G., A.H. and A.L.S. conceived of the measurements. A.L.S. fabricated the device. A.W.B., A.H. and A.L.S. performed transport measurements. A.W.B. and A.H. performed numerical simulations. A.W.B. wrote the manuscript with input from all other authors. K.W. and T.T. grew the bulk hBN crystals.

## Additional information

**Competing interests:** The authors declare no competing financial interests.

