## [Peer Review File · Nature Communications]

Reviewers' Comments:

Reviewer #1 (Remarks to the Author)

The Authors present a method for injecting highly-collimated electron beams in graphene, based on a double pinhole chamber that removes uncollimated trajectories.

The method is demonstrated experimentally to be effective, producing narrow beams with high transmission probability, and the experimental results are convincingly supported by theoretical computations.

In the pinhole chamber the numerical simulations are performed in a purely quantum framework (in order to account for diffraction effects) while, for the whole Hall-bar-like device, a simple semiclassical model (based on trajectories) is used. Yet, the semiclassical simulations give surprisingly good results and provide a useful tool for the interpretation of the experimental results.

To my knowledge, the method and the results exposed in this manuscript are new.

The proposed technique could be, potentially, very important for a technology aimed at exploiting the peculiar properties of chiral electrons in graphene.

My conclusion is that the manuscript should be accepted for publication on Nature Communications.

I suggest that some points could be clarified a little bit, before submitting a final version.

1) Fitting the simulation parameters "p_trans" and "p_scatter" to the data suggests that the asymmetric behavior of electrons and holes, evidenced in Figure 3, might be explained in terms of different scattering properties of the edges for electrons and holes. However, this is a rather unexpected behavior. Is it possible to provide a tentative explanation, or at least a comparison with other experiments?

2) Section 1 of Supplementary Information. Providing a general reference for eqs. (1) and (2) would help the not-expert reader.

3) Section 3 of Supplementary Information. In the conductance simulations, angles and amplitudes of injected plane waves are chosen at random. However, apparently, the energy of the plane waves is not fixed, so Authors should provide some details about the distribution of energies.

4) Section 3 of Supplementary Information. The reflective boundary is obtained by setting equal to 0 the u component of the wave function: setting equal to 0 the v component would be equivalent?

Finally, a short list of "proofreading" corrections:

1) Caption of Supplementary Figure 2: e)Same  e) Same

2) Caption of Supplementary Figure 4: hallbar  hall-bar ("Hall-bar", capitalized, would be even better, also in the main text)

2) Caption of Supplementary Figure 5: integration  Integration

Reviewer #2 (Remarks to the Author)

The authors realised a scheme for creating collimated electron beams in graphene. The analysis of their data is simple but agrees well with the observations.

The proposed absorptive collimator is interesting and will surely be useful for next-generation experiments in graphene, especially because of the relative simplicity of its realisation. Even though an alternative proposal already exists (PRL 118, 066801 (2017)) and should clearly be referred to, the present manuscript reports on an experimental realisation and is based on a different concept. Because of this, I recommend the manuscript for publication, provided the authors further adequately address the following issues.

1) The theoretical analysis is clearly below the standards of the experimental part. I do not see this as a critical shortcoming of the paper, whose main merits lie in the experiment itself, but a few points could be further clarified/commented upon.

1a) The word "semiclassical" is often used a bit recklessly, considering the mesoscopic context. When the authors talk of "semiclassical theory" in the conclusions, or about "semiclassical trajectories" here and there, they actually mean a theory that considers purely classical trajectories and neglects any sort of interference between (Feynman) paths, which is not what "semiclassical theory" is -- see e.g. the review R. A. Jalabert, Scholarpedia 11 (1), 30946 and refs. therein. A purely classical analysis is a fair starting point, and it seems to work rather well in this case, but it should not be presented as something that it is not. In particular, do the authors have any arguments as to why classical reasoning works so well in their phase-coherent sample? This could be very important when considering different geometries, as suggested in the conclusions ('...absorptive collimation in high-mobility graphene devices can be predictably and robustly applied in a variety of geometries...')

1b) Supp. Info Sec. 3 about diffraction: It would be helpful, especially for less-experienced readers, to explicitly mention that the staggered lattice approach cures fermion doubling problems. Also, it is not clear to me how the authors' calculations compare to standard non-equilibrium Green's function ones: Is the effect of their averaging procedure over amplitudes and angles comparable to that of describing injection via a lead self-energy?

2) Supp. Info, Sec. 4: What is "C3"? It seems not to be defined anywhere.

3) Supp. Info, Sec. 5: "When they hit an edge, the appropriate behavior-scatter, specularly reflect, refract, transmit (...) control parameters p_{scatter} (...) and p_{transmit} ". Shouldn't there be a control parameter for "refract" too?

Reviewer #3 (Remarks to the Author)

The authors propose an approach to obtain a not only narrow but also collimated electron beams by using both pinhole aperture structure. The grounded aperture design can avoid the background conductance originating from the stray electrons by removing them. It is also shown that narrow collimated beams can be used to characterize the edge of graphene devices. The experimental measurement and the theoretical simulation show a good agreement in this manuscript. The design in this work is technologically significant and has a great application potential in 2D electron systems.

When a narrow slit is used, the diffraction effect of electrons should be first considered. The authors owe the offset of n_0 to the diffraction in the main text and give a brief discussion about the diffraction effect when fitting the experimental data in Supplementary. Another fact is that the outgoing beam from the pinhole slits does not diverge at a large angle due to the diffraction, making a good collimation. This is important, but the authors should explain it by the comparison between the Fermi wavelength and the slit width in physics. In addition, the narrow electron beam claimed by the authors also should be defined by seeing whether its waist width is close to the Fermi wavelength.

Both the working efficiency of the design and the strength of the collimated beam depend on the transmission probability of electrons through the narrow slits. According to Fig.2b in the main text, the resulting collimated beam have the maximum current of about 2.8 nA. The authors should clarify whether such a collimated beam is enough strong to bear the burden expected by researchers in graphene.

Response to Reviewer #1:

1) Electron-hole asymmetry

Fitting the simulation parameters "p_trans" and "p_scatter" to the data suggests that the asymmetric behavior of electrons and holes, evidenced in Figure 3, might be explained in terms of different scattering properties of the edges for electrons and holes. However, this is a rather unexpected behavior. Is it possible to provide a tentative explanation, or at least a comparison with other experiments?

We thank the reviewer for this insightful question; indeed, a significant difference between scattering behavior for electrons and holes is not immediately intuitive. Regarding p_trans, the probability of transmission into a contact, asymmetry is commonly observed in experiments due to the work function of the contact metal. However, typically reported transport measurements (e.g. four-terminal R_{xx} and R_{xy}) are not sensitive to contact resistance, so the apparent symmetry in much of the literature does not reflect the asymmetry of contact resistance.

For p_scatter, the probability of diffuse scattering at an edge: etched edges of graphene are typically rough on the nanometer scale, and have generally been assumed to be highly scattering. If electrons never reach the physical edge, reflection could be more specular. In our group's recent work, we found that gating the edges of a device to form a PN junction parallel to the edge improves the specularly of reflection (likely due to total internal reflection from the junction). With this in mind, our working theory is that the etching process may leave some line charge that electron-dopes the edge. Thus in the electron-doped regime, there will be no PN junctions at the edges and electrons should reach the physical device edge and scatter. In the hole-doped regime, there will tend to be edge PN junctions, leading to smoother specular reflection.

As the reviewer suggests, we now provide the following tentative explanation:

"This asymmetry may occur due to finite edge doping that induces smooth electrostatic edge barriers³⁰ in the p-doped regime." This reference [30] is new to the manuscript: [30] "Evidence of the role of contacts on the observed electron-hole asymmetry in graphene", *Physical Review B* **78**, 121402(R) (2008) [new last sentence added to end of 1st paragraph on page 8 of manuscript]

2) Reference for 2D Landauer-Büttiker formalism

Section 1 of Supplementary Information. Providing a general reference for eqs. (1) and (2) would help the not-expert reader.

We have now incorporated a reference to “Ryndyk, D. A. in *Theory of Quantum Transport at Nanoscale* 17–54 (Springer International Publishing, 2016)” as suggested.

3) Clarification on energy conservation in quantum simulations

Section 3 of Supplementary Information. In the conductance simulations, angles and amplitudes of injected plane waves are chosen at random. However, apparently, the energy of the plane waves is not fixed, so Authors should provide some details about the distribution of energies.

We thank the reviewer for highlighting this important point. We in fact do preserve plane-wave energy in a given calculation of conductance and only vary it to simulate different electron densities. In response to the comment, we are correcting an error in one sentence that may alleviate a source of confusion:

We wrote “...we couple in a time-varying complex potential at the edge...” while we intended to say “...we couple in a time-varying complex wave amplitude at the edge...” This change has been made in the updated manuscript (2nd to last sentence, 3rd paragraph of Supplementary Note 3)

Also, we now make the following modification:

We added the phrase “at a fixed energy” into the following sentence: “In order to calculate total conductance of the collimator, we iteratively sum the plane-wave solutions at a fixed energy with random amplitudes...” (1st sentence, 4th paragraph of Supplementary Note 3)

4) Boundary conditions in quantum simulations

Section 3 of Supplementary Information. The reflective boundary is obtained by setting equal to 0 the u component of the wave function: setting equal to 0 the v component would be equivalent?

The reviewer’s intuition here is correct. Because we employ a staggered grid in our simulations, it is important that our boundaries not inadvertently cut a unit cell, which would effectively create point-like scatterers. The wave equation itself treats the u and v sublattices identically, however we adopt a master-slave approach where the boundary conditions are explicitly set for the u-lattice. To clarify this point, we have now added: “(setting boundary conditions on u instead of v is arbitrary; the reverse selection would be equivalent)”. (2nd sentence, 3rd paragraph in Supplementary Note 3)

5) Proofreading corrections

1) *Caption of Supplementary Figure 2: e)Same  e) Same*

2) *Caption of Supplementary Figure 4: hallbar  hall-bar ("Hall-bar", capitalized, would be even better, also in the main text)*

3) *Caption of Supplementary Figure 5: integration  Integration*

We thank the reviewer for the careful reading of our SI and for noting these corrections. We have now made all suggested changes in the manuscript.

Response to Reviewer #2:

1) Additional reference

...an alternative proposal already exists (PRL 118, 066801 (2017)) and should clearly be referred to...

We thank the reviewer for the suggestion of including a reference to this recent, elegant theoretical proposal. We have now included this reference in the first paragraph.

2) Use of the term "semiclassical" and validity of classical reasoning

The word "semiclassical" is often used a bit recklessly, considering the mesoscopic context. When the authors talk of "semiclassical theory" in the conclusions, or about "semiclassical trajectories" here and there, they actually mean a theory that considers purely classical trajectories and neglects any sort of interference ...

...do the authors have any arguments as to why classical reasoning works so well in their phase-coherent sample? This could be very important when considering different geometries, as suggested in the conclusions ('...absorptive collimation in high-mobility graphene devices can be predictably and robustly applied in a variety of geometries...') or perhaps different transport regimes.

We thank the reviewer for this important clarification and the subsequent question. Given that the collimators are a very "open" ballistic system, the distinction between semi-classical and purely classical treatments is likely minimal in practice, but it is important that we clarify the approach we use. In response, we have now changed "semi-classical" to "classical ballistic" in all cases where appropriate.

Regarding phase-coherence: In all of our measurements we ground at least one large contact to the bulk of the device, and check that the contacts have reasonable transmission probabilities. This condition is particularly well-satisfied in the electron-doped regime where the classical ballistic treatment works

best. Consequently, electrons rarely make a full circuit of our device and thus have no opportunity to interfere substantially if we restrict our consideration to classical paths.

In response to this important point, we have now modified the first sentence of the last paragraph on page 7 to be “To validate this understanding and quantitatively determine the degree of specularly, we next carry out device-scale simulations of ballistic trajectories—treating electrons as classical point-like particles is warranted given that the Fermi wavelength is in most cases much smaller than geometric features in our device and given that most trajectories are captured in ohmic contacts before having a chance to interfere.”

3) Details and clarifications on quantum simulations

Supp. Info Sec. 3 about diffraction: It would be helpful, especially for less-experienced readers, to explicitly mention that the staggered lattice approach cures fermion doubling problems. Also, it is not clear to me how the authors' calculations compare to standard non-equilibrium Green's function ones: Is the effect of their averaging procedure over amplitudes and angles comparable to that of describing injection via a lead self-energy?

The reviewer makes an excellent suggestion to make more direct comparisons to standard theoretical treatment. Our understanding is that solving for single particle wavefunctions (as is effectively done here) is formally equivalent to implementing the non-equilibrium Green's function approach in cases where there is no scattering and in the linear response limit, but we agree that the equivalence was not made manifest in the SI.

The Green's function approach typically posits semi-infinite leads that are coupled to the finite scattering region of interest. In this case, there are separable transverse modes that are either propagating or evanescent in the lead; the number of propagating transverse modes sets the maximum transmitted current through the scattering region. Our approach is somewhat unorthodox in implementation (with some similarity to previous work on non-adiabatic transport through QPCs). Given this, it is important that we ensure that transverse modes are evenly driven/populated as is true in the semi-infinite lead for the non-equilibrium Green's function treatment.

To that end, we have included a section in the SI labeled “Comparison of simulation to existing methods” discussion of the SI that numerically shows that our approach of picking random superpositions of incoming plane-wave states ultimately excites all propagating transverse modes equivalently.

We have now also included an explicit parenthetical statement “(which solves the fermion doubling problem)” (1st sentence, 3rd paragraph, Supplementary Note 3)

4) Proofreading

Supp. Info, Sec. 4: What is "C3"? It seems not to be defined anywhere.

We thank the reviewer for the detailed reading and feedback here. This label is in error: it should be "F3" to reflect the injecting collimators' filter contact, and we have made that correction in the updated manuscript.

5) Refraction control parameter

Supp. Info, Sec. 5: "When they hit an edge, the appropriate behavior-scatter, specularly reflect, refract, transmit (...) control parameters ρ_{scatter} (...) and ρ_{transmit} ". Shouldn't there be a control parameter for "refract" too?

While we mentioned the possibility of refracting in the description and we implemented it in our simulation code, we chose to neglect its contribution since the effective refractive index should be nearly constant throughout the bulk of the device at the doping levels that we operate in for much of our work. There may be some refractive behavior at the device boundaries, but this can be parametrized as a combination of reflection and scattering given the proximity to the edge, so a new tuning parameter is not required.

In consideration of the reviewer's helpful suggestion, we eliminated the word "refract" from the above sentence, and add the statement: "Refractive behavior is not considered in this analysis given that the electron density is expected to be uniform over the bulk of the device." as the 3rd to last sentence in the first paragraph of the "Ballistic simulations" SI section.

Response to Reviewer #3:

1) Details on diffraction

When a narrow slit is used, the diffraction effect of electrons should be first considered. The authors owe the offset of n_0 to the diffraction in the main text and give a brief discussion about the diffraction effect when fitting the experimental data in Supplementary. Another fact is that the outgoing beam from the pinhole slits does not diverge at a large angle due to the diffraction, making a good collimation. This is important, but the authors should explain it by the comparison between the Fermi wavelength and the slit width in physics. In addition, the narrow electron beam claimed by the authors also should be defined by seeing whether its waist width is close to the Fermi wavelength.

This important point the reviewer flags clearly deserves discussion. We were very interested in the role of diffraction and found its apparent ability to explain the conductance behavior worth highlighting in

Fig 2a in the main text. However, we also hope to maintain the accessibility of this work to a non-expert audience—particularly in relation to the discussion of Fig. 1 in the main text. With this in mind, we now have included an explicit statement of the Fermi wavelength in caption of Fig. 1a, and begin Supplementary Note 1 with the sentence: “For most electron densities covered in the main text, electrons can be viewed as classical point-like particles whose momentum distribution is determined by the band-structure and Fermi-Dirac statistics.”

We have also modified the last paragraph in Supplementary Note 3:

This effect is likely due to the reflective boundary conditions of the first pinhole, as well as diffraction throughout the collimator. At density n_0 , the Fermi wavelength $\lambda_f = 89 \text{ nm}$ is an appreciable fraction of the width of the slits; under these conditions, the beam diffracts significantly off of itself, constraining the available spatial modes that fit between slits. While a detailed analysis of the mode shapes is beyond the scope of this work, it is worth noting that the Fraunhofer diffraction off a single slit with this low density only has $\theta_{FWHM} \approx 17^\circ$, which is still narrower than the classical angular distribution discussed above. This means that even under modest doping, diffraction does not appreciably broaden the collimated beam.

The remainder of the preexisting paragraph is continued as a separate paragraph “A way to phenomenologically parametrize the net effect of diffraction and interference is to assert ...”

2) Strength of collimated beams

Both the working efficiency of the design and the strength of the collimated beam depend on the transmission probability of electrons through the narrow slits. According to Fig.2b in the main text, the resulting collimated beam have the maximum current of about 2.8 nA. The authors should clarify whether such a collimated beam is enough strong to bear the burden expected by researchers in graphene.

The reviewer brings up an excellent point that the specific currents we measure and report (<3 nA) in the doubly collimated configuration are somewhat small from a pragmatic standpoint. The doubly-collimated collected current is proportional to the injected current, which is why we use a ratio for the y-axis of Fig. 2b. The material limitation on the graphene device’s current density is nowhere near saturated, so we expect higher currents can be used. Here we chose our injecting current to preserve the effective electron temperature. To that end, we have now included the following inequality to make the reasoning of this choice explicit: $I_{\text{source}} = 50 \text{ nA} < \frac{k_B T}{e R_{\text{source}}}$. We have now also included a Supplementary Note labeled “Estimate of attainable collimator currents” that states:

“In the main text's discussion pertaining to Fig. 2, we sourced $I_{\text{source}} = 50 \text{ nA} < \frac{k_B T}{e R_{\text{source}}}$, where $R_{\text{source}} \sim 1 \text{ k}\Omega$ is the resistance from the source to all other contacts. The choice to operate in this low injection current regime ensures that the applied bias is below the thermal bandwidth

($k_B T = 140 \mu\text{eV}$) at our measurement temperature of $T = 1.6 \text{ K}$, and that the measurement stays well within the linear response regime.

In our measurements, we did not probe the maximum current that could be sourced while maintaining collimation; however this is a relevant consideration for technological applications. With this in mind, ohmic contacts of the size we use can readily pass 20x more current than in our measurement without degradation. If we were to increase injected current to this level ($1 \mu\text{A}$), we would not expect substantial changes in the collimated beam shape: the thermal bandwidth would be significantly increased but would remain below 30 K. Recent work has indicated that the mean free path at this energy scale would be $\sim 10 \mu\text{m}$ [5]; this is large compared with the width of our device ($2 \mu\text{m}$), which means that ballistic transport should still dominate. Considering this, we anticipate that the relatively low current in the doubly-collimated configuration could be enhanced to $\sim 60 \text{ nA}$, without appreciable loss of collimation.

Even higher energy scales/electron temperatures could bring the electron system into the hydrodynamic regime. It is not yet clear what this would do to beam collimation.

Reviewers' Comments:

Reviewer #1:

Remarks to the Author:

I confirm my previous appraisal on the quality of this manuscript. The proposed revisions have been properly considered and some important clarifications have been given where needed. As a consequence, I would suggest the publication of the manuscript in the present form.

Reviewer #2:

Remarks to the Author:

The authors adequately addressed all issues raised in the previous round. I recommend the paper for publication in Nat. Commun.

Reviewer #3:

Remarks to the Author:

The authors have addressed all my concerns. I recommend the paper to be published in Nature Communications